# Implementing HRD Testing in Routine Clinical Practice on Patients with Primary High-Grade Advanced Ovarian Cancer

**DOI:** 10.3390/cancers15030818

**Published:** 2023-01-29

**Authors:** Florian Heitz, Beyhan Ataseven, Claudia Staniczok, Carsten Denkert, Kerstin Rhiem, Eric Hahnen, Sebastian Heikaus, Malak Moubarak, Julia Welz, Timoleon Dagres, Vasilios Vrentas, Mareike Bommert, Stephanie Schneider, Nicole Concin, Philipp Harter

**Affiliations:** 1Klinik für Gynäkologie und Gynäkologische Onkologie, Evangelische Kliniken Essen Mitte, 45136 Essen, Germany; 2Klinik für Gynäkologie mit dem Center für Onkologische Operative Therapie, Charité Campus, Virchowklinikum, Charité—Universitätsmedizin Berlin, Freie Universität Berlin, Humboldt Universität zu Berlin, Berlin Institut of Health, 13353 Berlin, Germany; 3Department of Obstetrics and Gynecology, University Hospital, LMU Munich, 81675 Munich, Germany; 4Institut für Pathologie, Universitätsklinik Marburg, Philipps-Universität Marburg, 35043 Marburg, Germany; 5Center for Hereditary Breast and Ovarian Cancer, Center for Integrated Oncology (CIO), Medical Faculty, University Hospital Cologne, 50937 Cologne, Germany; 6Zentrum für Pathologie Evangelische Kliniken Essen Mitte, 45136 Essen, Germany

**Keywords:** primary high-grade advanced ovarian cancer, homologue recombination deficiency, *BRCA1/2*, PARP inhibitor, tumor testing

## Abstract

**Simple Summary:**

Testing for a homologous recombination deficiency (HRD) in primary high-grade ovarian cancer is crucial for recommending the appropriate therapy. Patients with a BRCA germline mutation are defined as harboring an HRD deficiency. However, there is a group of approximately 20% of patients without a BRCA mutation which harbor an HRD deficiency based on tumor genomic analyses. In the current research, we wanted to share the experience of a high-volume tertiary cancer center with implementing central HRD testing—in another institution—to reduce doubts about logistic issues and to the highlight critical aspects that hinder HRD testing. We showed that the results of HRD testing became available in a timely manner for the therapy decision. However, it was important to obtain as much tumor tissue as possible during the first diagnosis—before any other intervention—as the tumor quantity and quality were critical for HRD testing. Additional BRCA germline testing further reduced the failures of HRD testing and should additionally be routinely conducted.

**Abstract:**

The chemotherapy backbone for patients with high-grade advanced epithelial ovarian cancer (HG-AOC) is carboplatin and paclitaxel followed by a maintenance therapy either with bevacizumab, with a PARP inhibitor, or with a combination of both, which is defined by the presence of a homologous recombination deficiency (HRD) and by the *BRCA1/2* status. This study included patients with a primary diagnosis of HG-AOC treated between December 2019 and December 2021. The HRD status was measured using the Myriad myChoice^®^ test on all the patients with an indication for tumor HRD testing. Germline testing was conducted on all the patients using the TruRisk^®^ panel as recommended by the national guidelines. HRD testing was requested for 190 patients, and, for 163 patients (85.8%), an HRD test result was available. An HRD test result could not be reported in 27 patients due to an insufficient tumor yield. The median time that it took to receive the HRD test results was 37 days (range of 8–97). In total, an HRD was present in 44.7% (73/163) of the patients based on a GIS ≥ 42 in 42.9% of the patients and based on a tumor *BRCA1/2* mutation in 3 cases (all with a GIS < 42). The germline testing results were available for 148 patients, and, in 18 patients (12.2%), a deleterious germline mutation was detected. Of the 27 patients without sufficient HRD testing, *BRCA1/2* germline testing results were available for 19 patients (70.4%), and a deleterious germline mutation was detected in 2 patients (7.4%). The implementation of HRD testing is feasible, and the results become available for treatment decisions in a timely manner for most patients. The prerequisite for HRD testing with the Myriad myChoice^®^ test is a sufficient amount of tumor tissue. The cotesting of HRD and *BRCA1/2* germline testing should be aimed for in order to enable optimal and timely treatment decisions on maintenance therapy as well as to test patients on whom the HRD test will not be evaluable.

## 1. Introduction

Primary ovarian cancer is the second most common malignancy of the female genital tract and the most lethal gynecologic cancer in developed countries [1,2]. The columns of treatment include primary surgery followed by chemotherapy, which consists of carboplatin and paclitaxel [3]. In the GOG 218 and ICON 7 trials, it was shown that the addition of bevacizumab to the chemotherapy and its use as a maintenance therapy—for a total of 15 months—was associated with a prolongation of the progression-free survival (PFS) [4,5]. Treatment with poly(ADP-ribose) polymerase inhibitors (PARPis) has been evaluated for nearly 10 years in recurrent ovarian cancer—first in patients with a *BRCA1/2* mutation and later in patients without a *BRCA1/2* mutation [6,7,8,9]. Due to their high efficacy in the recurrent disease setting, phase 3 trials were conducted in the primary setting. The general principle of the below-described trials was maintenance therapy. Patients with high-grade ovarian tumors were required to have undergone platinum-based chemotherapy and must have responded with at least partial remission. Following this, the patients were randomized into groups of treatment with a PARPi or with the placebo (olaparib for 2 years, niraparib for 3 years, and rucaparib for 2 years). The SOLO-1 study demonstrated that the patients with a *BRCA1/2* mutation who received maintenance therapy with olaparib had a significantly longer PFS than the patients treated with the placebo [10]. The PRIMA study included patients with and without a *BRCA1/2* mutation and showed a significant prolongation of the PFS (8.2 vs. 13.8 months; *p* < 0.001) in favor of the patients treated with niraparib [11]. The PAOLA1 study also included patients with and without *BRCA1/2* mutations. However, the patients were required to receive bevacizumab in addition to the chemotherapy, which was also to be given as a further maintenance therapy for 12 months after the completion of the chemotherapy. In the intention-to-treat group, the PFS was also significantly prolonged for patients additionally treated with olaparib (16.6 vs. 22.1 months; *p* < 0.0001) [12]. Mutations in the *BRCA1/2* genes can lead to limitations in the effectiveness of the “deoxyribonucleic acid repair metabolic pathways” (DNA damage repair (DDR) pathways). These are involved in the detection and repair of DNA damage to maintain the genomic integrity of the cell [13,14,15]. One of the six DDR pathways described so far is the so-called “homologous recombination pathway”. Tumors with a homologous recombination deficiency (HRD) show a higher proportion of chromosomal instability. In addition, DNA double-strand breaks (caused, for example, by treatment with platinum-containing agents but also by PARP inhibitors) cannot be repaired sufficiently [16,17,18]. The main proteins active in the HRD pathway are *BRCA1* and *BRCA2*. However, in addition, several other genes are active in this pathway, and the failure of these proteins can equally lead to an HRD. Myriad’s HRD assay was established in advance of the above studies and was used in both the PAOLA1 and PRIMA studies. The subgroup analyses in both the PRIMA and PAOLA-1 studies showed that patients with HRD and/or *BRCA1/2* mutations had a strong response to PARPi treatment, as did patients without HRD or *BRCA* mutations. The PRIMA study demonstrated a PFS benefit with PARPi even in patients without an HRD, but the PAOLA1 study did not. Accordingly, niraparib was approved for all patients, regardless of their *BRCA1/2* or HRD status, but olaparib was approved only for patients with a *BRCA1/2* mutation or an HRD in combination with bevacizumab. In addition to a high-grade histology and a response to the chemotherapy, either a *BRCA* mutation or an HRD must be present. The last one can be evaluated using the Myriad myChoice^®^ panel.

Due to the strong prognostic and predictive value of the HRD and BRCA test results for individual patients, a European expert consensus recommended the use of BRCA and HRD testing for recently diagnosed patients with advanced ovarian cancer [19]. The aim of the present study was to evaluate the implementation of the—currently only centrally available—Myriad myChoice^®^ test for use in clinical practice and to determine its concordance between germline *BRCA1/2* testing and HRD testing in affected patients.

## 2. Materials and Methods

Patients treated between December 2019 and December 2021 in the tertiary cancer center “Department of Gynecology and Gynecologic Oncology” of Kliniken Essen-Mitte with advanced (≥FIGO IIIA) high-grade ovarian cancer (HG-AOC) were included. The Myriad myChoice^®^ test was implemented and validated at the cooperating Institute of Pathology, Philipps-Universität Marburg, Germany, and was officially approved for testing in a collaboration with Myriad on decentral HRD testing. Archival formalin-fixed paraffin-embedded (FFPE) tissue from patients with primary diagnosis of ovarian cancer outside of our department was requested from external pathology departments whenever quality and/or quantity of tumor tissue was not sufficient for HRD testing. In addition to the genomic instability score (GIS), the HRD test also reports variants in *BRCA1*, *BRCA2*, *ATM*, *PALB2*, *BARD1*, *RAD51C*, *RAD51B*, *RAD54L*, *BRIP1*, and *CDK12*. Indication for germline testing was based on the guidelines of the Center of Hereditary Breast and Ovarian Cancer (HBOC) at the University of Cologne, Germany. Counseling was performed by gynecologic oncologists according to HBOC’s standard. A detailed description of the process was described earlier [20]. Neither germline nor tumor testing was conducted at our department, we solely extracted and documented the results of the analyses in our database. The germline analysis was performed using the TruRisk^®^ gene panel, which covers BRCA1/2 genes and *ATM*, *CDH1*, *CHEK2*, *MLH1*, *MSH2*, *MSH6*, *PMS2*, *PALB2*, *RAD51C*, *RAD51D*, *BARD1*, *BRIP1*, and *TP53*. Comparisons of frequencies were analyzed based on variable categories with chi-square test or Fisher’s test. A *p*-value < 0.05 was defined to be significant. Statistical analyses were conducted with SPSS Version 23.0 (IBM Corporation, New York, NY, USA) software.

## 3. Results

The HRD test was indicated for 196 patients, which were treated in the designated interval. For 6 patients, the test was abrogated (see Figure 1), leading to 190 patients of whom the HRD test was finally ordered (Figure 1). However, not enough tumor tissue for HRD testing was available for 27 patients. Data for *BRCA1/2* germline test results were available for 168 patients (88.4%); 5 patients refused testing; 3 patients did not fulfill the inclusion criteria for cost coverage of their insurance companies due to advanced age; 8 patients’ insurance companies refused testing due to other reasons; and 6 patients’ tests were not conducted, as a *BRCA1/2* mutation/HRD was shown to be present in the tumor.

The patients’ characteristics are displayed in Table 1. The majority were diagnosed to be in FIGO stages IIIC/IV (82.1%) and underwent primary debulking surgery (PDS; 71.1%), with 95.8% of the patients having high-grade serous differentiated cancers.

The median duration between the HRD test order and the receipt of the test result was 37 days. The duration for the first two samples in IVQ 2019 was above the average with a median of 55 days, but there were no statistical differences between the respective intervals during the implementation phase (*p* = 0.48) (Figure 2a).

The median duration between the genetic counseling/blood draw and the receipt of the test result was 23 days. There was a significant variability in the median turnaround times for germline testing (min. of 17.5 days and max. of 29 days) within the respective intervals during the implementation phase of the HRD test (*p* < 0.0001) with no significant impact on clinical management (Figure 2b). Figure 1 displays the feasibility of HRD testing and the frequencies and causes for the HRD test failures. Finally, sufficient HRD test results with reports of the GI-score were obtained for 163 patients but were not obtained for 27 patients (14.2%). However, in 5 of the latter 27 patients’ somatic mutation analyses, three pathogenic somatic *BRCA1* mutations, one *CDK12* mutation, and one variant of unknown significance in *RAD54L* were identified.

The reason for the nonsufficient HRD test results was an insufficient tumor quantity for all the samples, and the reasons, therefore, were classified as follows: primary debulking surgery/biopsy performed for diagnosis with an insufficient amount of tumor tissue (N = 5/N = 4), an outside diagnosis with an insufficient amount of tumor tissue from the outside and no interval debulking surgery (IDS) (N = 6), and an insufficient amount of tumor tissue from the outside and an insufficient amount of tumor from the IDS (N = 12). Figure 3 displays that not enough tumor tissue was gathered in 14/56 patients (25%) in the first observational period, while that frequency was reduced to 13/113 patients (11.5%) in the second observational period. Due to the learning experience that tumor sampling is critical—mainly before neoadjuvant chemotherapy—we gathered more tissue during primary diagnosis. Therefore, the frequency of tumor-related reasons for an insufficient tumor yield seemed to increase (PDS with an insufficient amount of tumor tissue), while the frequency of an insufficient amount of tumor tissue during the primary diagnosis before neoadjuvant chemotherapy decreased during the implementation phase.

Of the 163 patients with an available GI-score, an HRD (cutoff ≥ 42) was detected in 70 patients (42.9% of the patients with an HRD test result). In an additional 3 patients (1.8% of the patients with an HRD test result), an HRD was defined due to a somatic *BRCA1* mutation with GI-scores of 40, 28, and 26, leading to a positive HRD score in 44.7% of the patients.

In 24 of the 70 patients with an available GI-score and a defined HRD, a somatic *BRCA1/2* mutation was detected (34.3%); thus, an HRD without a *BRCA1/2* mutation was detected in 46 patients (65.7%). Of the 163 patients with a sufficient HRD test, *BRCA1/2* germline testing results were available for 149 patients (91.4%), and, in 18 patients (12.2%), pathological germline mutations were detected. Of the 27 patients without a sufficient HRD test, *BRCA1/2* germline testing results were available for 19 patients (70.4%), and, in 2 patients (7.4%), pathological germline mutations were detected (see Figure 4). In 30 tumors (15.8%) of the patients undergoing HRD testing (N = 190), somatic *BRCA1/2* mutations were also identified, which were also found through germline testing in 14 patients (N = 62.5%).

## 4. Discussion

The present study profoundly showed that the implementation of decentral HRD testing is feasible despite it being centrally performed at a pathology institute in another state. Although there was some delay in its implementation on the first patients, the general implementation phase was short, and the general turnaround time for receiving the HRD results was acceptably long for treatment decision. Nevertheless, the most critical issue for conducting HRD testing is the amount of tumor tissue that must be available. A considerably high number of patients did not receive a sufficient HRD test result due to a shortage of available tumor tissue, and some of those patients might not have received optimal treatment due to that. HRD testing is important for the approval-compliant prescription of therapy with bevacizumab and olaparib. Tumors from patients who are eligible for treatment with bevacizumab and who have high-grade carcinomas should undergo HRD testing.

Neither patients with early nor low-grade ovarian cancer are approved for maintenance therapy with bevacizumab and olaparib. There is undoubtful evidence that an HRD is of prognostic importance in patients with ovarian cancer [21,22]. However, patients who are not scheduled for bevacizumab maintenance therapy do not require mandatory HRD testing, as an HRD alone does not qualify them for maintenance therapy with olaparib. In the subgroup analyses of the PRIMA study, the patients with HRD and somatic *BRCA1/2* mutations were shown to have a strong response to niraparib maintenance therapy alone (HR: 0.4 (95% CI of 0.27, 0.62)), as were the patients who had evidence of an HRD but no somatic *BRCA1/2* mutation (HR: 0.5 (95% CI of 0.31, 0.83)). In addition, the patients without an HRD still showed a PFS benefit with niraparib (HR: 0.68 (95% CI of 0.49, 0.94)) [11]. Accordingly, maintenance therapy with niraparib was approved for all patients with advanced high-grade tumors who have responded to platinum-containing chemotherapy. In contrast, the subgroup analyses of the PAOLA-1 study showed that maintenance therapy with bevacizumab and olaparib resulted in a median improvement in the PFS of 19.5 months in the patients with HRD-positive tumors (including somatic *BRCA1/2* mutations) compared with the placebo-controlled group. The patients with HRD-positive tumors (without somatic *BRCA1/2* mutations) showed a median PFS improvement of 11.5 months, which was also statistically significant. Admittedly, the patients who were HRD-negative did not show an improvement in their PFS with the addition of olaparib to bevacizumab [12]. Based on these results, olaparib in combination with bevacizumab was approved for patients with high-grade carcinomas who have responded to platinum-containing chemotherapy or who have “no evidence of disease” and have either a *BRCA1/2* mutation or an HRD, exclusively.

One unanswered question which remains is the definite treatment decision for patients with HRD-negative tumors. Those patients might be treated, on the one hand, with bevacizumab during chemotherapy and further treated for 12 months as a maintenance therapy. On the other hand, those patients might be treated with the chemotherapy alone and receive niraparib as a maintenance therapy for 3 years. As depicted in Figure 5, a “natural” cycle could be to ask at the beginning of the chemotherapy whether there is a need for bevacizumab (e.g., high tumor burden, tumor-related liquid burden such as ascites, or pleural effusion). The other question should be whether there are any contraindications for bevacizumab’s usage (e.g., bowl leakages, chronic inflammatory bowel disease, or uncontrolled hypertension). The AGO study group just recently launched the randomized phase III trial AGO-OVAR 28 (NCT05009082) in which patients with advanced ovarian cancer are randomized into carboplatin/paclitaxel or carboplatin/paclitaxel and bevacizumab treatment groups (for chemotherapy and as a maintenance therapy for 12 months further), and all the patients will receive niraparib as a maintenance therapy. This study will also give us more insight into the optimal treatment approach for patients with HRD-negative tumors. The main results of the analysis described here relate to two important logistic aspects that should be considered. On the one hand, HRD testing should be initiated as early as possible to have the result of the test promptly available for the definitive planning of the maintenance therapy. The results in our study were available after a median of 37 days, which was fine for patients who underwent a primary surgery and then received six cycles of chemotherapy every 3 weeks (an approximately 126-day duration) and which was in line with previously reported central genomic analyses [23]. If patients underwent an interval debulking surgery and then received three more cycles of chemotherapy and then the test results took > 90 days—as it did three times in our study—then the results would not be available for the definitive planning of the maintenance therapy, which is suboptimal. One of the most important drawbacks for a faster receipt of the HRD results in our institution was the refund of the test in the German medical system. Tests are not covered within the Diagnosis-Related Groups (DRGs); thus, tests cannot be ordered until the patient is treated at an outpatient level. Therefore, it is strongly advised that the costs of these test should be included in the ovarian cancer DRGs. It is important to note, however, that, over the observed implementation period, the long time to receive the HRD results in the beginning were minimized (Figure 2a). Another important logistic aspect that was shown in the present analysis was the high rate of nonmeaningful HRD findings—in the present work, this occurred for 14.2% of patients—which was a little lower than the 18% reported in the PAOLA study [23]. The nonmeaningful HRD findings were exclusively due to insufficient available tumor materials. Although 6 of the 27 patients without meaningful HRD findings already had insufficient tumor tissue available during the primary surgery, in 2/3 of the patients, the primary specimen obtained through laparoscopy or biopsy was insufficient in size. The expectation that sufficient tumor tissue will be found in the interval debulking surgery (IDS)—when the biopsy to confirm the diagnosis may have been very small—to perform HRD testing is often misleading since, by this time, large parts of the tumors are often already in remission; as in our collective, where we could no longer find sufficient tumor tissue in 15 patients during their IDS. It is true that, especially in patients for whom a diagnosis has not yet been confirmed, the basic question is which underlying disease is present. However, the yield of the tumor tissue, to confirm the diagnosis, should be sufficiently high because the effectiveness and the approval status of the maintenance therapy to be given later can be significantly dependent on the HRD status. The amount of the sufficient yield of the tumor tissue at diagnosis is also an important aspect for reducing turnaround times for HRD testing. In some cases, other FFPE tumor blocks were asked for in order to repeat testing if the first test was not successful, thus prolonging the time until the HRD test was available.

Another aspect that was highlighted in the present study was the correlation of HRD/somatic BRCA testing and panel testing. Due to the high rate of nonmeaningful HRD findings, germline testing cannot be omitted. A total of 2 of the corresponding 27 patients were found to have a BRCA1/2 germline mutation who would have been deprived of an effective therapy without germline testing. On the other hand, 37.5% of the patients with a somatic *BRCA1/2* mutation did not have a corresponding germline mutation, so somatic testing alone would assume a much too high hereditary burden, which would impose an unnecessary burden on the patients and families concerned. The latter issue of the discordance rates between *BRCA1/2* tumors and germline testing is known. In the PAOLA trial, 29/114 patients (25.4%) and, in the AGO TR-1 study, 31/393 patients (7.9%) with a *BRCA1/2* mutation in the tumor had no mutation in the germline, respectively [23,24]. Since the latter studies used the same BRCA assay for germline and tumor testing and since we used different tests, the higher rate of discordance in our study might be explained. However, the performance of the Myriad myChoice^®^ test in our analysis was consistent with the previously published data. The described rates of HRD positivity, but also of somatic *BRCA1/2* mutations, were confirmed. A total of 48.3% of the successfully screened tumors were HRD-positive in our study, and this occurred for 50.8% of the tumors in the PRIMA study and for 48% of the tumors in the PAOLA-1 study [6,7]. The rate of somatic *BRCA1/2* mutations as the reason for HRD positivity was also in the same range: 37% in our study, 30.4% in the PRIMA study, and 29% in the PAOLA-1 study. The high concordance of the results underlines the validity and applicability of the Myriad myChoice^®^ test in a decentralized setting as recently described by Denkert et al. [12]. However, there are strong academic efforts that are underway to validate other HRD tests in order to replace the Myriad myChoice^®^ test and to make testing available in an even more decentralized setting [25,26,27].

## 5. Conclusions

In summary, the present work showed that the implementation of HRD testing in a decentral pathological department is feasible and that results are available for most patients for treatment decisions in a timely manner. However, the prerequisite for HRD testing is enough tumor tissue, which should be taken early—at first diagnosis of the disease—as it is rather unlikely that enough tumor tissue will be available later after chemotherapy initiation. The cotesting of HRD and *BRCA1/2* germline testing should be also aimed for in order to enable optimal and timely treatment decision on the maintenance therapy for patients in whom the HRD test will not be evaluable. Further research is demanded to evaluate technologies which need a lower amount of tumor tissue for HRD testing.

## Figures and Tables

**Figure 1 cancers-15-00818-f001:**
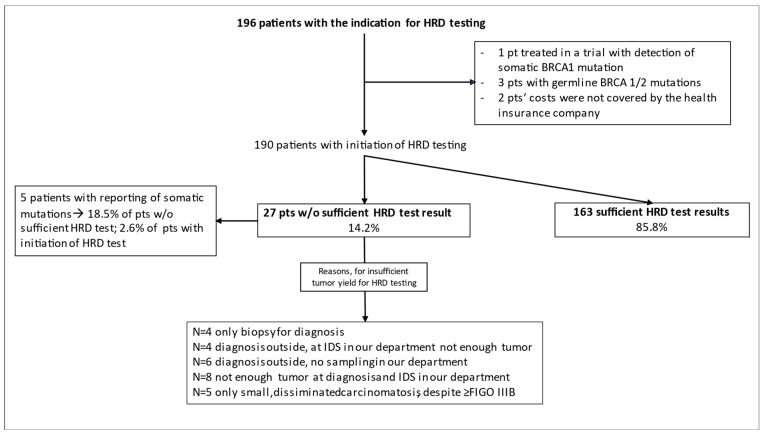
Flow chart of patients with the indication for HRD testing and final results. HRD represents homologous recombination deficiency; IDS represents interval debulking surgery.

**Figure 2 cancers-15-00818-f002:**
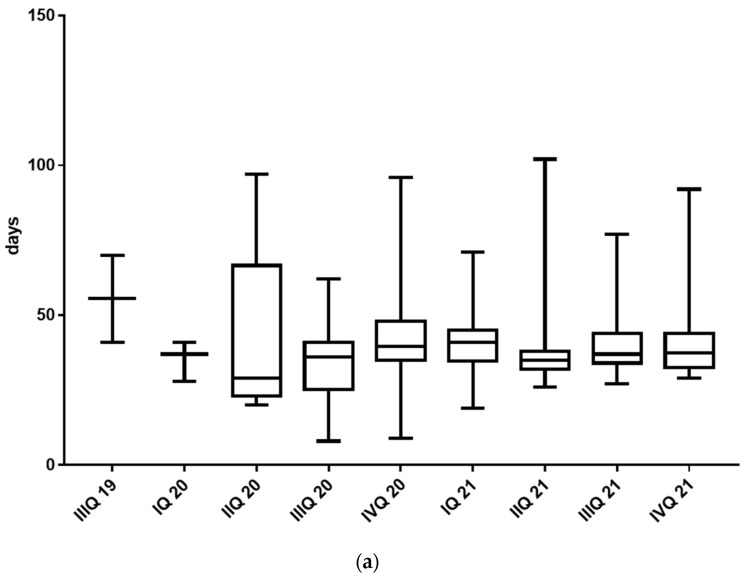
(**a**) Time between ordering and reporting the results of the HRD test in different time intervals during the implementation phase. (**b**) Time between ordering and reporting the results of the established germline test in different time intervals during the HRD test implementation phase.

**Figure 3 cancers-15-00818-f003:**
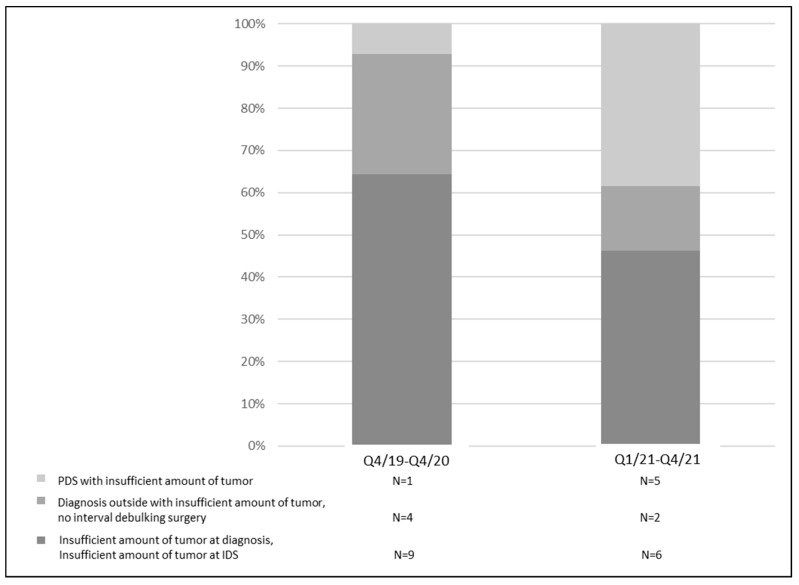
Reasons for insufficient amount of tumor tissue, leading to insufficient HRD testing during the implementation phase; Q4/19–Q4/20 N = 56, and Q1/21–Q4/21 N = 113.

**Figure 4 cancers-15-00818-f004:**
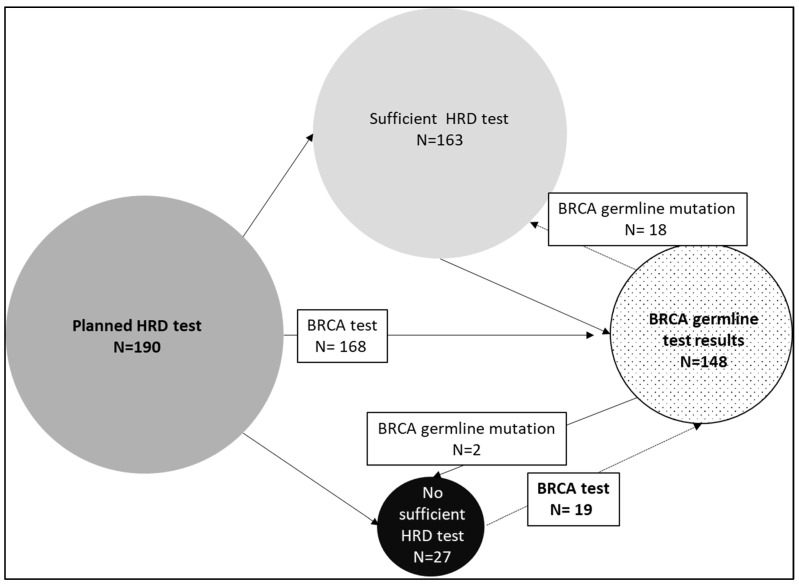
Description of HRD test results in combination with *BRCA1/2* germline test results. HRD represents homologous recombination deficiency; BRCA represents breast cancer.

**Figure 5 cancers-15-00818-f005:**
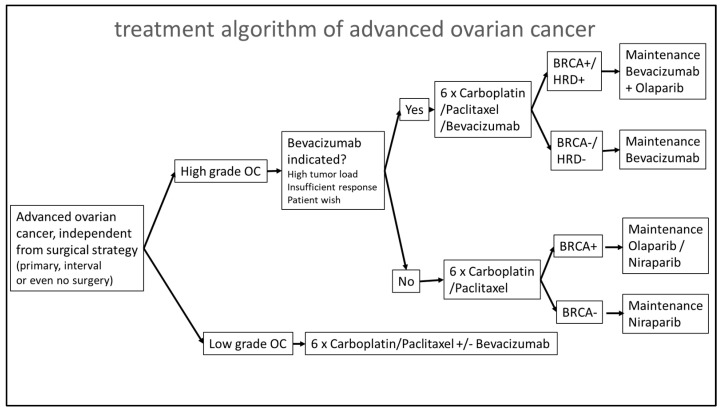
Treatment algorithm for maintenance therapy of patients with advanced ovarian cancer based on clinical and molecular biological parameter.

**Table 1 cancers-15-00818-t001:** Patients’ characteristics. FIGO—Federation of Gynecology and Obstetrics staging system; ECOG-PS—Eastern Cooperative Oncology Group performance score; HRD—homologous recombination deficiency; and §—result with report of genomic instability score (GI-score).

Parameter	N = 190 (%)
Age (Median; Min.–Max.)	62; 23–88
FIGO IIIA IIIB IIIC IV	10 (5.3) 24 (12.6) 64 (33.7) 92 (48.4)
ECOG-PS 0 > 0	171 (90.0) 19 (10.0)
Surgery Primary debulking surgery Interval debulking surgery No surgery	135 (71.1) 40 (21.1) 15 (7.8)
Histology High-grade serous High-grade endometrioid Clear cell Mucinous destructive/infiltrative	182 (95.8) 2 (1.1) 5 (2.6) 1 (0.5)
*BRCA1/2* germline testing Yes No	168 (88.4) 22 (11.6)
Sufficient ^§^ HRD test result Yes No	163 (85.8) 27 (14.2)

^§^ result with reporting of Genomic instability-score (GI-score).

## Data Availability

The data presented in this study are available on request from the corresponding author. The data are not publicly available due to privacy reasons.

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
