# Peer review of "Implementing HRD Testing in Routine Clinical Practice on Patients with Primary High-Grade Advanced Ovarian Cancer"

_cancers, 2023, doi:10.3390/cancers15030818_

Round 1
Reviewer 1 Report
Good and important work.
Author Response
Thank you very much for your contending comment!
Reviewer 2 Report
Many thanks for letting me read this manuscript. It reports the feasibility of performing HRD testing in the clinical practice of a German specialised institution.
Overall, the manuscript is well written although, it requires an english revision in some parts.
The topic is of moderate interest since in the current practice HRD, as stated also by authors should be made available to all the patients requiring bevacizumab combination in first line in order to allow for "PAOLA-1" combo. In this respect, the authors should deeper discuss about the potential factors that can be improved in order to have a even earlier result.
Author Response
Thank you very much for this comment. We have included the follwing sentence to the discussion, which reflects one potential optimisation aspect. " One of the most important drawbacks for a faster receipt of the HRD results in our institution is the refund of the test in the German medical system. Tests are not covered within the Diagnosis Related Groups (DRG), thus tests can be ordered not until the patient is treated on an outpatient level. Therefore it is strongly advised, that costs of these test should be included to the ovarian cancer DRGs."
A further aspect was included:"The amount of a sufficient yield of tumor tissue at diagnosis is also an important aspect to reduce turnaround times for HRD testing. In some cases other FFPE tumor blocks were asked for to repeat testing, if the first test was not successful, thus prolonging time until the HRD test was available."
Reviewer 3 Report
This is a very nice real world study about the implementation and institutional experience with HRD and germline BRCA testing in ovarian cancer. The topic is sound and the number of included patients is relevant. The length of the paper is adequate and the manuscript well written. The article adds relevant data about the somatic and germline testing experience in this large center which are very interesting and an important as a reference for the readers in the field.
Minor Points
1) The Myriad MyChoice(R) HRD testing was used in the present study. The authors discuss the limitation of insufficient tumor tissue (at laparoscopy or biopsy) as a reason for inadequate HRD results. The authors prompt the necessity of improved test methods possibly requiring less tissue and the conclusion is adequate. As another limitation regarding the test sensitivity it could also be mentioned in the discussion that "HRD" tumors might be missed by the current test (the authors mentioned the improved PFS of "HRD neg." for Niraparib in PRIMA).
2) The discussion about the choice of the maintenance treatment based on HRD testing etc. is important and well done. It would be nice to add a decisional diagram as an additional figure to illustrate the decision making and recommendations.
3) Discussion: What is the author's recommendation and usual practice in patients HRD neg. who would qualify for both, either maintenance with Niraparib or Bevacizumab?
4) Fig2a/b: please translate "Tage"
Author Response
Minor Points
1) The Myriad MyChoice(R) HRD testing was used in the present study. The authors discuss the limitation of insufficient tumor tissue (at laparoscopy or biopsy) as a reason for inadequate HRD results. The authors prompt the necessity of improved test methods possibly requiring less tissue and the conclusion is adequate. As another limitation regarding the test sensitivity it could also be mentioned in the discussion that "HRD" tumors might be missed by the current test (the authors mentioned the improved PFS of "HRD neg." for Niraparib in PRIMA).--> thank you very much
2) The discussion about the choice of the maintenance treatment based on HRD testing etc. is important and well done. It would be nice to add a decisional diagram as an additional figure to illustrate the decision making and recommendations. --> thank you very much for this comment. HAve included figure 5. to display our tratment algorithm.
3) Discussion: What is the author's recommendation and usual practice in patients HRD neg. who would qualify for both, either maintenance with Niraparib or Bevacizumab? --> added to the discussion:"One unanswered question which remains is the definite treatment decision for patients with HRD negative tumors. Those patients might be treated on the one hand with bevacizumab during chemotherapy and further on for 12 months as maintenance therapy. On the other hand, those patients might be treated chemotherapy alone and receive niraparib as maintenance therapy for 3 years. As depicted in figure 5, a “natural” cycle could be to ask at the beginning of chemotherapy, if there is a need for bevacizumab (e.g. high tumor burden, tumor-related liquid burden like ascites, or pleura effusions). The other question should be, if there are any contraindications for bevacizumab usage (e.g. bowl-leakages, chronic inflammatory bowel disease, uncontrolled hypertension). The AGO study group just recently launches the randomized phase III trial AGO-OVAR 28 (NCT05009082) in which patients with advanced ovarian cancer are randomized to carboplatin/ paclitaxel or carboplatin/ paclitaxel and bevacizumab (during chemotherapy and as maintenance therapy for further 12 months), and all patients will receive niraparib as maintenance therapy. This study will give us more insight into the optimal treatment approach, also of patients with HRD negative tumors."
4) Fig2a/b: please translate "Tage"--> thank you very much for this tip. Figures were changed and new figures were included